# WHAT DOES GPT STORE IN ITS MLP WEIGHTS?
# A CASE STUDY OF LONG-RANGE DEPENDENCIES

## ABSTRACT

Large Language Models such as GPT are able to recall factual information about the world that they have learnt during training. This information must be stored in the model weights yet there is much we do not know about exactly what information is stored, where it is located and how it is retrieved. In this paper, we test and develop existing theories about information storage and retrieval through the example of bracketed sentences. We show that, in the case of recognizing brackets, where a model must learn during training to associate matching opening and closing brackets, very early multi-layer perceptron (MLP) layers in the source position are responsible for this association. This supports existing work on the importance of MLP layers as key-value memory stores (Meng et al., 2023) and a potential hierarchy of roles within transformers, whereby early layers are responsible for storing and retrieving lower level information compared to more abstract information which is stored in later layers (Geva et al., 2021).

## 1 INTRODUCTION

Transformer-based language models (Vaswani et al., 2017; Brown et al., 2020), have demonstrated impressive natural language understanding, but their inner workings remain largely opaque. Gaining insight into these models is challenging due to their complex, densely connected architecture and high-dimensional feature space. As these models are already deployed in important real-world applications (Zhang et al., 2022), understanding and anticipating their behavior is critical. Some argue interpretability is key for the safe deployment of advanced ML systems (Hendrycks et al., 2022). The emerging field of **mechanistic interpretability** aims to reverse engineer model computation into human-understandable components (Elhage et al., 2021; Meng et al., 2023). By uncovering underlying mechanisms, we can better predict out-of-distribution behaviors (Foote et al., 2023; Mu & Andreas, 2020), identify errors (Nixon et al., 2020), understand emergent behaviors (Nanda et al., 2023; Wortsman et al., 2019), and more.

In this work, we use circuit analysis (Räuker et al., 2023) to mechanistically understand how GPT-2, a generative large language model, performs a simple linguistic task: identifying closing brackets. We identify a subgraph of the model responsible for this behavior. To discover the circuit, we employ **activation patching** (Meng et al., 2023), a method that iteratively traces the contributions of individual components of the model to the final output distribution. We supplement this with embedding projections, attention analysis, and linear probes to understand the role of each component.

Among the various tasks that LLMs are capable of, we focus on understanding the simple process of how they **identify and match closing brackets**, as it provides a tangible insight into their underlying mechanisms in detecting long-range dependencies. The proposed circuit (Figure 1) consists of early MLP layers in the source (opening bracket) position that are responsible for retrieval of the matching token pair and attention heads in early layers which are responsible for moving this information to the final position. Our analysis demonstrates that LLMs, such as GPT2, store information in their MLP layers and that low-level linguistic information, such as bracket associations and word pairings, are stored in very early layers. This is consistent with the findings of Geva et al. (2021) who propose that progressive MLP layers are responsible for storing increasingly complex information. Contrary to Geva et al. (2021) and Meng et al. (2023), in the case of brackets, we identify a single MLP layer rather than a range of layers that is responsible for this behavior. We also observe that the key and value vectors representing different vectors do not have a privileged basis and so do not align with

the model's natural basis, as implied by Geva et al. (2021). An additional finding is that the residual output directly after this MLP layer cannot be used to directly predict the correct token, indicating that the retrieved information lies initially in a dormant direction and is only later activated.

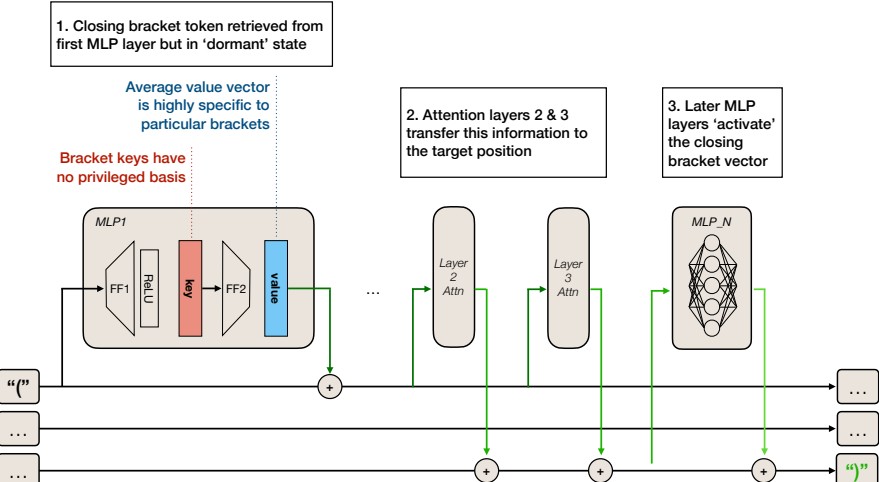

Figure 1: The matching closed bracket token is stored in the first MLP layer in the source position. The vector representing the closed bracket, despite the presence of the ReLU, does not align with a dimension of the model. The vector retrieved from this layer is highly specific to a particular bracket and patching in the average vector for a specific bracket type is enough to predict the corresponding output bracket regardless of the open bracket in the original input. The resulting residual activation cannot be used to directly predict a closed bracket, indicating that the retrieved information lies "dormant" within the model until activated by later MLP layers in the target position.

## 2 BACKGROUND

We experiment with $N$-layer transformer models with a vocabulary size $V$. The model takes an input sequence $\boldsymbol{x} = x_1, \ldots, x_p$ where each $x_i \in \{1, \ldots, V\}$. Tokens are mapped to $d_e$-dimensional embeddings by selecting the $x_i$-th column of $E \in \mathbb{R}^{d_e \times V}$, the embedding matrix. Each $d_e$-dimensional embedding then passes through $N$ layers, consisting of multi-head attention and MLP blocks.

Every transformer layer contains a residual connection, so models can be viewed as having a residual "stream" that each block reads from and writes to (Elhage et al., 2021). In the multi-head attention block, the $d_e$ dimension embedding gets split into $n_{head}$ streams of dimension $d_{head} = d_e/n_{head}$. Attention distributions are computed independently for each head via the expression $\text{softmax}\left(\frac{QK^\top}{\sqrt{d_{head}}}\right)V$, where $Q, K, V \in \mathbb{R}^{d_e \times d_{head}}$. The MLP block applies two linear transformations separated by a non-linearity (ReLU) independently across sequence positions. After the $N$ layers, the final linear layer $U \in \mathbb{R}^{V \times d_e}$ projects the residual stream representations to vocabulary space to get logits. The residual stream is important because information is read from and written to a single vector space across layers. Projecting activations to vocabulary space can elicit emergent behavior (Belrose et al., 2023). This technique contributes to an understanding of when information is learned within the network.

We analyze how information flows through the transformer architecture using *activation patching* in a similar way to Meng et al. (2023). We use it in the context of brackets and simple word pairings as opposed to more abstract factual associations. This technique makes targeted edits to network activations during the forward pass and observes the effect on output logits. These edits involve replacing specific activations, for example the output of a specific MLP layer in a particular sequence position, with corrupted values. This could involve adding noise to the existing activations or replacing them entirely with different values, either averaged values across many different inputs or taken

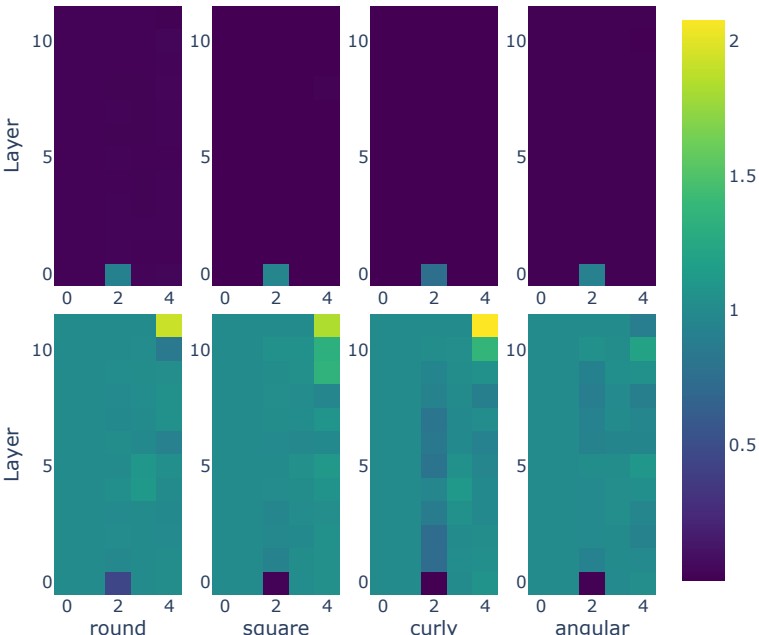

Figure 2: Relative difference in closed bracket output probability when patching the MLP output of a specific layer at a given position compared to the original prompt with an open bracket in the second position. The top row shows positive patching where the input is a random sequence and values are patched from activations saved from the original prompt. The bottom row shows negative patching, where the input is the original sequence and values are patched from a previous run involving a random sequence. Each column represents a different bracket type.

from a previous forward pass using a different prompt. During one forward pass, multiple edits can be made in different locations to investigate how combinations of components operate within the network. The difference in output logits with and without model edits provides information about how those specific components of the network contribute to the task. We describe activation patching in more detail in §3.

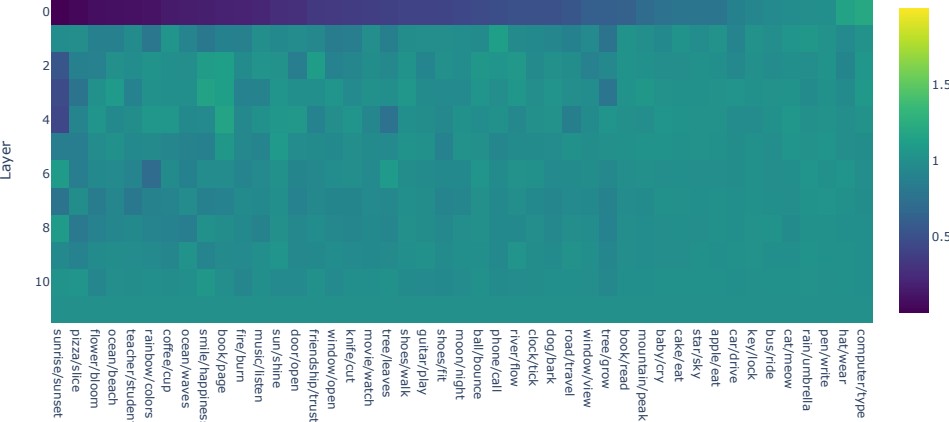

Figure 3: Negative patching of common single token pairs. The color represents the change in probability of predicting the second token after negative patching relative to the original prompt containing the first token. The vertical axis represents results for patching specific MLP outputs in the source position with the activation from an equivalent run with a random sequence token. The horizontal axis represents different token pairings, sorted by relative effect of the the first MLP layer.

## 3 METHODOLOGY

We focus on bracket completion task as it represents a direct symbolic dependency that must be learned during training. The model needs to track opening and closing brackets across arbitrary distances, requiring properly handling state information. Bracketing naturally decomposes into state tracking, memory lookup, and contextual cues, enabling analysis of how these mechanisms compose. It relies on both prompt context and learned knowledge, elucidating their interplay. The self-contained structure of brackets makes the full circuit tractable to study as an example of handling linguistic dependencies. Brackets form a "shallow" syntactic pattern likely using early model layers, contrasting factual knowledge in later layers.

Take for example the sentence fragment *John gave the bottle (containing milk*. As a self-contained linguistic structure, bracketing provides a tractable phenomenon to study the emergence of long-range dependencies between tokens within transformers. We propose bracket completion involves three key processes:

1. **State**: is the current position inside or outside a bracketed sequence? Information from the opening bracket position must be transferred to the closed bracket position so that the model can understand that it is currently within a bracketed sequence. How is this information transferred and processed within the network?

2. **Memory**: what is the appropriate closing token to go with the opening bracket? The model must store information about the closed bracket symbol within the model weights. Where is this information stored? How is it retrieved using information about the source token?

3. **Context**: has the phrase reached a point where the close bracket is needed? To accurately predict a close bracket, the model must combine information about the current phrase and whether it has reached a conclusion. How do these different bits of contextual information get combined?

This toy problem is similar to Indirect Object Identification (Wang et al., 2022) in that it is composed of multiple sub-tasks that must interact correctly together to perform the overall task. We believe that more examples explaining circuits of this complexity will be helpful in laying the foundation for exploring more advanced model behaviours. We do not offer a complete explanation of the process and instead focus on the second point: where information matching pairs of brackets is stored and how this information is retrieved. We therefore only study the influence of a single open bracket token and do not present findings concerning bracket completeness and the effect of a closing bracket on subsequent bracket prediction probability (point 1). We average across multiple random sequences to remove the contribution of broader context to the bracket prediction task (point 3).

To reduce complexity and isolate important behaviour, we focused on short sequences ($n = 5$) of random tokens. We found longer sequences to display broadly similar behavior but needlessly increase the complexity of analysis by providing alternative paths through which information can flow forward through the network from early to late positions. We found indirect attention, the flow of information from source to target position across multiple layers via intermediate positions, to be very prevalent and made it difficult to analyse the precise role of individual attention layers. As a result, we found it helpful and more tractable to keep sequences short.

We investigate four separate types of brackets: round "(", ")", square "[", "]", curly "{", "}" and angular "<", ">". Experiments were carried out on GPT2-small and, where stated, GPT2-XL[1]. Unless stated otherwise, the open bracket token[2] is located at the second position (zero indexed) and we observe the probability of outputting a closed bracket token in the fourth position. The indices for tokens at all other positions were selected randomly from a uniform distribution in the range 256 to 5256 in order to choose common tokens but not single bytes. Unless otherwise stated, probabilities are the result of averaging 100 different randomly sampled input sequences. We use random tokens in order to isolate the effect of the open bracket and reduce the noise caused by other meaningful

---

[1]The model weights were downloaded from HuggingFace via the TransformerLens library. GPT2-small consists of 12 layers (85M parameters) and GPT2-XL consists of 48 layers (1.5B parameters).

[2]The GPT2 tokenizer uses byte pair encoding and there are multiple tokens that represent open and closed brackets. We uniformly sample between each type for the opening bracket and sum the probabilities of each closing bracket. Full details of which tokens were included are included in B

signals in the case of logical sequences. We tested our findings separately on a selection of hand crafted sentences to check that the effects discovered do generalise to meaningful sentences.

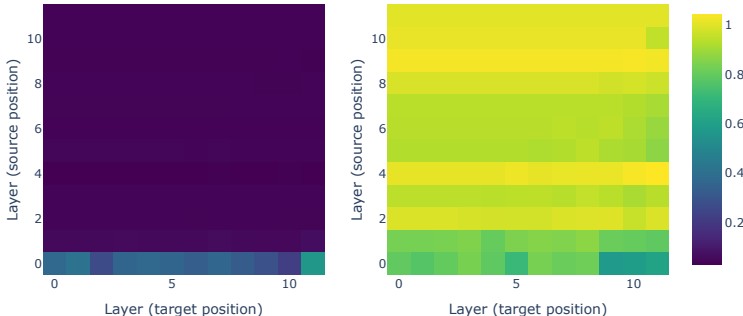

Figure 4: Source and target MLP ablation for round brackets. Two initial forward passes are completed, one with a baseline random sequence and another with the same sequence but an opening bracket in the source position. The baseline and bracketed activations for these respective forward passes are saved and used for negative and positive patching in the following runs. The left plot starts with the random sequence as input and applies positive patching of the bracketed activations at a specific MLP layer in the source position (y-axis) and repatching of the baseline activations in subsequent MLP layers of the source position, followed by negative patching of the original baseline activations at a specific MLP layer in the target position (x-axis). The right plot shows the inverse of this: starting with the bracketed sequence and applying negative patching in the source position (with repatching of the true activations in subsequent source MLP layers) and positive patching in the target position. Sequence length is 4 with the open bracket in the position 3.

**Patching**  Our main tool to test the role of the different layers in predicting token dependencies is patching. This tool, formalised by Wang et al. (2022), works by running the neural network in two modes. The first mode is a vanilla one, running the input prompt as usual, and then calculating the logit probabilities of the next token prediction. For example, running the prompt "Bill Gates founded" and testing the logit for the token "Microsoft". In addition, the circuit is also run in "corrupt mode", in which internal representations of the model run on a new prompt are replaced by representations from another run on a different prompt. For example, for the prompt "Steve Jobs founded", we might replace the internal representations of some layer at the position of the token "founded" with some internal representations from the Bill Gates prompt, and then continue to run the model as usual. Patching is done only on part of the representations - a complete patching, where all layers and positions are replaced, will lead to an identical run as the original prompt.

Within the realm of patching, there are two primary modes - negative and positive patching, sometimes referred to as direct and indirect effects (Pearl, 2001). Negative patching involves selectively corrupting certain portions of the internal representation. Essentially, this introduces "noise" or "disturbances" into the model's internal inference process. By observing how the model's output changes in response to this perturbation, one can glean insights into which parts of the representation are crucial for a given output. If a model's output drastically changes after a particular patch, it suggests that the patched section was integral to the original output. For instance, consider the prompt "The sun rises in the". A model might predict "east" as the next token. Now, if we negatively patch the representation associated with the word "rises", and the model suddenly predicts "west", it indicates that the perturbed representation was crucial in guiding the original prediction.

Positive patching is the opposite of negative patching. Instead of corrupting or disturbing a portion of the representation, positive patching involves enhancing or reinforcing certain aspects. This is achieved by borrowing and overlaying representations from related prompts. If the model's output becomes more confident or shifts in a particular direction following a positive patch, it suggests that the overlaid representation carries significant contextual information related to the output. As an example, let's take the two prompts "Bill Gates founded" and "Steve Jobs founded". If we overlay the representations from "Bill Gates" onto the prompt related to Steve Jobs, and the model suddenly becomes more confident in predicting "Microsoft", it shows that the overlaid representation was influential in making that prediction.

Whilst these two methods are mechanically identical, except for switching around the sequences, they tell us something different about the contribution of a specific part of the network: positive patching identifies if a subgraph of a network is *sufficient* for performing a particular task whereas negative patching identifies if the same subgraph is *necessary* for performing that task.

# 4 RESULTS

We turn next to describe the main results of our study. We find that the retrieval of the matching bracket token is carried out in the first layer in the source position (§4.1). The information retrieved from this layer is specific for a given bracket type but cannot be directly interpreted as the correct token (§4.2).

## 4.1 EARLY MLP LAYERS ARE NECESSARY FOR ACCURATE BRACKET PREDICTION

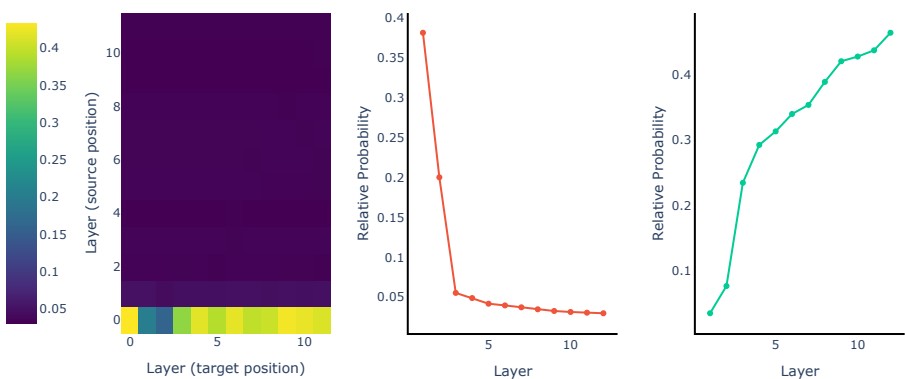

Figure 5: The left hand plot is a similar experiment to the one shown in the left hand plot of Figure 4. But with attention blocks in the target position patched rather than MLP blocks. The other two plots start with a randomised baseline sequence but apply positive patching to the output of the first MLP layer in the source position. They measure the relative probability compared to the original bracketed sequence when applying negative patching to a set of attention layer outputs in the final position. The central plot shows the effect of patching all attention layers less than or equal to the current layer and the right plot shows the effect of patching all layers greater than the current layer.

Figure 2 shows that, in GPT2-small, the first MLP layer in the source position is critical for the accurate prediction of closed brackets, for all four bracket types. It is noticeable that, across all layers and positions, it is the only MLP layer which causes any significant increase in output bracket probability. In the case of negative patching, the final layers in the target position lead to an increase in output probability, indicating that these layers normally have a moderating effect.

There are a number of possible hypotheses for the role of the first MLP layer:

- It is important for all predictions and performs some, as yet unkown, general algorithmic task which is not specific for memory retrieval or the brackets task in particular.
- The matching token is retrieved from this MLP layer but the information for different types of bracket are stored in different parts of the layer and are retrieved independently.
- Its role is specific for paired token retrieval more generally, perhaps making the network aware that it needs to retrieve a matching token, but that the matching token is retrieved later in the network from other MLP layer(s).

Further investigation shows brackets are not the only token pairs which show a high dependency on the first MLP layer. The extent of this effect varies, as shown in Figure 3, but is consistent across a wide range of prompts. We take a variety of single token pairs that commonly appear together (e.g. sunrise and sunset) and perform negative patching to observe which MLP layers lead to a significant decrease in probability of outputting the matching token in the target position. Once again, for the vast majority of cases, it is the first layer in the source position which shows strong dependence and there are very few cases where any other MLP layers strongly inhibit the output.

It is not the case that the first layer is essential for all language tasks, as evidenced by the few cases in Figure 3 that do not rely on the first MLP (e.g. star and sky), but more generally by other, more complex pattern matching tasks (see Figure 8 and related work e.g. Meng et al. 2023) which tend to rely heavily on later layers.

One hypothesis for the importance of the first MLP layer is simply that it's the first layer and therefore has the largest downstream effects. To eliminate this as a possibility, the experiment described in Figure 2 was repeated but with the following modifications: (1) The original output of each subsequent layer is patched back into the residual layer so that it is only the direct effect of a specific MLP layer that can alter the information transferred to the final position, rather than any downstream effect is has on MLP layers in the same position. (2) Reducing the sequence length so that the open bracket directly preceeds the final position. This ensures that attention can only take place directly from final position to source position and excludes any composition through intermediate positions. (2) Patching attention and MLP layers in the final position. This tests the possibility that there are layers early in the network that are important in the final position for processing information from the source token and that this is why the first MLP layer in the source position is so important.

The results of this are shown in Figure 4 and confirm that the first MLP layer is important because of the computation it is able to perform rather than its location right at the start of the network. Even with repatching of subsequent MLP layers in the source position, the first MLP layer leads to a large increase in closed bracket probability. It is only later MLP layers in the target position that are important, which therefore do not rely on information being transferred to the target position early on in the network.

These experiments were repeated again but attention blocks rather than MLP blocks were patched in the final position (Figure 5). Here, early layer are of more importance than later layers. Notably, if the target position cannot attend to the source position during the first 3 attention layers, the probability of outputting a closed bracket token drops by a factor of ten. Whatever information is extracted from the first MLP layer in the source position is largely transferred to the target position via the first few attention layers in the network, even though it is much later MLP layers in the final position that appear to be responsible for processing this information.

A clear difference between our work and Meng et al. (2023) is that we find the very first MLP layers of GPT to be important for memory retrieval rather than early to mid layers. In §A we conclude that this is not due to prompt length and is likely due to the more foundational nature of the information being retrieved (Geva et al. 2021; ).

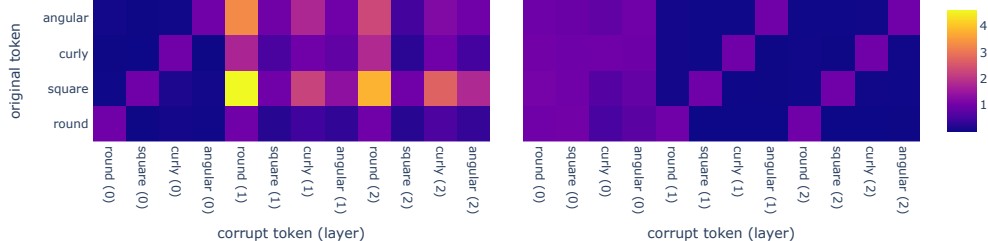

Figure 6: Activation swapping. Average MLP output activations were created by averaging across 100 samples for each bracket type and MLP layer. The top row shows the relative difference in probability of the original closed bracket token when patching an MLP layer with the average activation of another bracket type. The bottom row shows the same but for the probability of predicting the closed bracket that matches the incoming activation used to patch the MLP layer. Only layers 1 - 3 are shown here but the remaining 9 layers are very similar to layers 2 and 3.

## 4.2 INFORMATION RETRIEVED IS SPECIFIC BUT DORMANT

Language models can only store information they have learnt during training in their weights. In the case of brackets, knowledge of which closed bracket token is paired with which open bracket token is something that must be learnt during training and therefore stored in model weights. It has been shown in the previous section that there is only one critical layer (the first MLP layer in the source position) that this could feasibly apply to. This is the case for all types of brackets and also many

common word-based token pairings. This poses two possible hypotheses: (1) The matching token for all of these pairings is stored in different directions within this MLP layer. (2) The first MLP layer performs some other generic task, e.g. potentially indicating the general "importance" of the current position for future positions to subsequently attend to. The matching token is then stored elsewhere in a distributed fashion without being located in a specific layer.

Building on the concept of MLP layers as Key/Value memory retrieval systems (Geva et al., 2021), the final activation from the MLP block that is added to the residual stream can be thought of as a value vector representing the retrieved fact. Averaging the value vectors in the source position across a batch of sequences with the same open bracket token, we obtain an "average" value vector. The effect of replacing activations with the averaged activation from another token can then be measured in order to ascertain if they have the same functionality.

Figure 6 shows that the average activation vectors for each bracket type are highly uncorrelated. Swapping the first layer MLP output for the averaged activation from another bracket type dramatically reduces the probability of outputting the original bracket token and leads to a very high probability of outputting the new closing bracket token. This is the same for all bracket types. It is extremely noticeable that the first layer is the only layer which this applies to and swapping averaged activations into other layers does nothing to increase the probability of outputting the new closed bracket symbol and has an inconsistent effect on the original bracket probability.

It is worth noting that the output of the first MLP layer itself cannot be thought of as directly representing the matching closed bracket. It is increasingly clear that the vector added to the residual stream is crucial for predicting the closed bracket and, in some sense, must surely represent it. However, directly passing the residual stream activation after the first MLP layer through the final unembedding (as in LogitLens (Nostalgebraist, 2020)) does not lead to a significant increase in closed bracket prediction when the first MLP is corrupted with the bracketed activation. Indeed, it is only in much later layers (in both the source and target position) that the first MLP layer makes a significant difference (Figure 7).

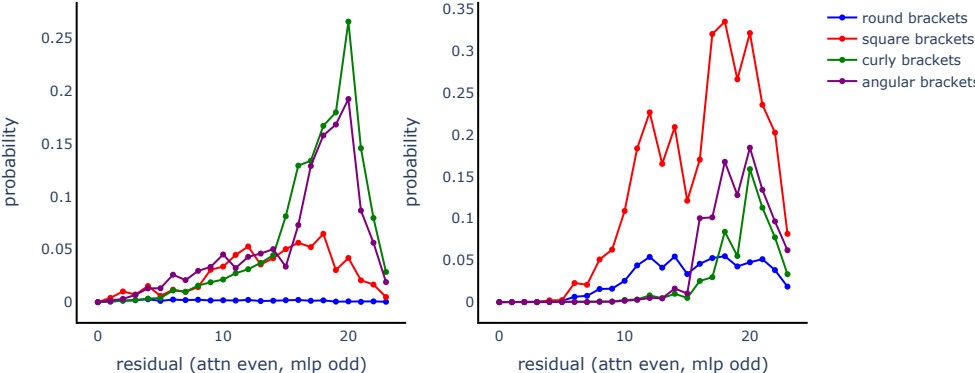

Figure 7: Starting with a randomized baseline sequence, the first MLP output in the source position is positively patched with the activation from a previous run with an open bracket. The residual activations at all points in the source (left) and target (right) positions are then mapped directly to logits by passing those activations directly the final layer norm and unembedding layer. We plot the probability difference of the correct closing bracket with and without patching the first MLP output.

## 5 RELATED WORK

There has been growing interest in mechanistically interpreting the inner workings of large language models like GPT. Prior work has explored techniques for analyzing how information flows through transformer networks. Räuker et al. (2023) propose circuit analysis to trace model computations into human-understandable components. They argue that decomposing model behavior into causal pathways can enhance interpretability. Our work similarly aims to elucidate mechanisms via circuit analysis on a concrete linguistic task.

Attention layers have received significant focus as a more interpretable component of transformers (Elhage et al., 2021). Geva et al. (2021) specifically highlight the potential of MLP layers as memory storage, framing them as key-value systems. Building on this, Meng et al. (2023) demonstrate editing MLP weights after training to alter factual associations stored in layers 6-9 of GPT-2 XL. We find factual recall utilizes different layers than our bracket task.

Wang et al. (2022) introduce activation patching to study emergence within transformer models. Meng et al. (2023) expand patching as a diagnostic technique to trace model computations. We adopt patching to identify components integral to bracket completion. Relatedly, Wang et al. (2022) trace indirect object resolution via patching attention pathways. Our analysis traces a distinct linguistic phenomenon. Belrose et al. (2023) show projecting intermediate activations to vocabulary space can reveal when information is learned within transformers.

We find bracket retrieval arises in early layers, but only later directly influencing predictions. This aligns with findings on redundant operations (McGrath et al., 2023; Pires et al., 2023) and dormant representations (Bolukbasi et al., 2021).

Overall, our work contributes additional evidence toward a hierarchical understanding of transformer learning (Geva et al., 2021), with early layers capturing shallow dependencies, like brackets, before later composition of deeper factual knowledge. Analyzing simple linguistic circuits clarifies base mechanisms enabling more complex behaviors.

## 6 Conclusion, Limitations and Future Work

Our analysis provides evidence that very early MLP layers in GPT-2 are responsible for storing and retrieving low-level linguistic dependencies like paired brackets. The first MLP layer contains distinct representations for different bracket types that can be swapped to alter predictions. This aligns with the theory that MLP layers form key-value storage systems. However, the "values" do not directly yield the matching token, suggesting the relevant representation may initially be dormant before later layer activation.

The prominence of early layers differs from factual knowledge tracing where middle layers are critical. This likely reflects meaningful differences between shallow linguistic versus deeper factual knowledge within transformers. Our findings support the hierarchical hypothesis which shows early layers capture shallow syntactic patterns before later composition of more complex representations.

In this paper we analyzed a syntactic dependency in a small model. It will be important to extend analysis to more complex structures and different models. Future work should trace additional pathways like tracking state information across layers and other examples such as nested brackets.

We believe exploring other local dependencies like quotes could reveal commonalities to be an interesting immediate future direction. Developing a more precise delineation between "shallow" and "deep" linguistic patterns learned across layers would strengthen theoretical hierarchical accounts.

## 7 Reproducibility Statement

To ensure our work is reproducible, we provide the full source code in the supplementary materials, as well as all necessary data and parameters and instructions to reproduce our experimental results.

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

## A    ADDITIONAL RESULTS: EARLY VS MID MLP LAYER IMPORTANCE

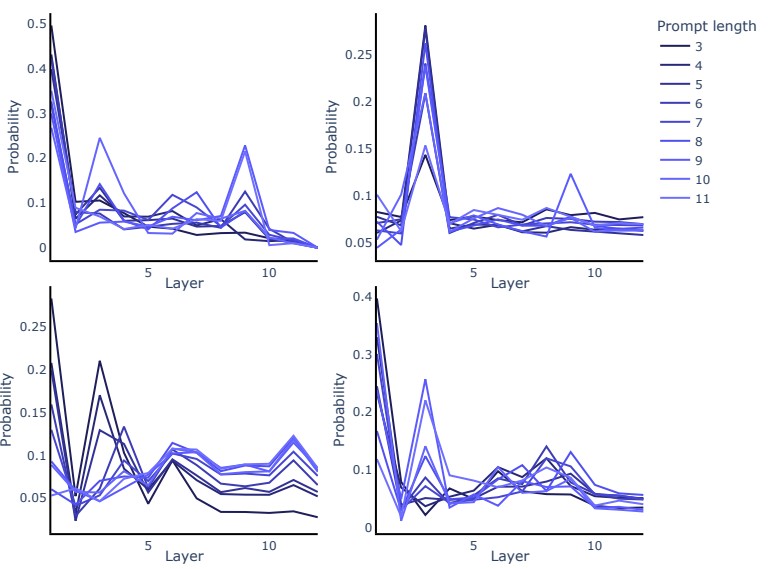

Figure 8: Patching with and without repatching. Clockwise from top left: negative patching with no repatching, negative patching with repatching, positive patching with repatching and positive patching with no repatching. Starting with a randomized initial prompt (positive patching) or the true prompt (negative patching) and then replacing the output of a specific MLP activation in the final prompt position with the activation from the alternative prompt. With repatching, all other MLP output activations in the same position are repatched with the original activations to mitigate indirect effects. Each line represents the average across all samples of a given prompt length. In the case of positive patching, the probability is measured relative to the baseline prompt, in the case of negative patching it is the probability relative to the correct prompt but subtracted from 1.

One striking difference between these results and previous explorations of factual recall in transformer MLP layers is the importance here of the very first layer. In Meng et al. (2023) it is middle layer MLP blocks which are responsible for storing factual associations. One obvious point of difference is that GPT2-small has far fewer layers (12) than GPT2-XL (48) but the difference is notable even when taking this into account.

We hypothesised that this could be a result of prompt length. In the case of a bracket, which can be represented with a single token, there is no need for composition between tokens across multiple positions in the prompt in order to form a key vector representing their combination. This composition likely takes place over numerous layers and so the network may learn to store associated values in later MLP layers as a result.

We tested this using the 1000 factual prompts used by Meng et al. (2023) for causal tracing. In order to remove any examples which the model definitely does not know, we accept only prompts where the probability of outputting the first token in the target is above a given threshold value set to 0.05. We also remove any prompts for which there are fewer than 10 examples of the same prompt length.

We analyze only the MLP outputs in the final prompt position (e.g. in "The Eiffel Tower is located in", the final prompt position would be the token "Tower") as Meng et al. (2023) show this position is the most important. We carried out experiments on both GPT2-small (individual layers, 493 examples) and GPT-XL (10 layer sliding window, 1063 examples). We report results for both positive and negative patching. For each of these cases, we examine results with and without repatching of subsequent MLP layers in the same position. Repatching reduces indirect effects and does not allow the patched MLP layers to alter the output probability by affecting other later MLP layers.

In Figure 8, we observe a similar pattern to Meng et al. (2023) in that early to mid layers in the final prompt position are of greatest importance and that late layers have almost no impact on output probability. We report that only in the case of negative patching without repatching is there a significant correlation between prompt length and average layer importance. However, this pattern is a result of very high dependence on early layers in the case of short prompts rather than the peak in importance of middle layers shifting slightly earlier. We therefore conclude that there is no clear correlation between prompt length and the location of important MLP layers within the network.

Instead, it is very likely the case that the difference in location of information storage is due to the type of information stored. In the case of more complex factual associations (Meng et al., 2023), information is stored in later MLP layers, whereas lower level linguistic features, such as brackets and simple word associations are stored in very early layers (Geva et al., 2021).

## B   MORE EXPERIMENTAL DETAILS

The GPT2 tokenizer uses byte pair encoding and there are multiple tokens that represent open and closed brackets. We isolate all instances of each bracket that occur within the first 3000 tokens:

- Round brackets: open: '(', '_(', closed: ')', ').', '),', '_)', ');', '.)', '):'
- Square brackets: open: '[', '_[', closed: ']', '_]'
- Curly brackets: open: '{', '_{', closed: '}', '_}'
- Angular brackets: open: '<', '_<', closed: '>', '_>'

We uniformly sample between each type for the opening bracket and sum the probabilities of each closing bracket.

## C   PSEUDOCODE

---

**Algorithm 1** Symbolic Sequence Predictions with Activation Patching

---

1: **Initialization:**
2: Load LLM model (e.g., GPT)                                                  ▷ Initialize model
3: Define bracket pairs                                                        ▷ Set bracket pairs

> **Step 1:**

       **Bracket Identification**
4: **for** each token in the sentence **do**
5:     **if** token is an opening bracket **then**
6:         Mark position and type                                   ▷ Tag open brackets
7:     **end if**
8: **end for**

> **Step 2:**

       **Activation Patching**
9: Cache activations                                                           ▷ Store activations
10: Get logit indices for responses                                           ▷ Identify response tokens
11: Extract uncorrupted log probs                                             ▷ Get true log probabilities
12: Patch residual stream                                                     ▷ Apply positive patching
13: Obtain corrupted logits with hooks                                        ▷ Apply negative patching
14: Extract corrupted log probs                                               ▷ Get corrupted log probabilities

> **Step 3:**

       **Contextual Information**
15: **for** each token in the sentence **do**
16:     **if** context suggests bracket end **then**
17:         Predict matching closing bracket                        ▷ Match open-close brackets
18:     **end if**
19: **end for**

> **Step 4:**

       **Finalize Predictions**
20: **for** each token in the sentence **do**
21:     **if** unmatched opening bracket **then**
22:         Insert predicted closing bracket                        ▷ Close open brackets
23:     **end if**
24: **end for**

> **Step 5:**

       **Output**
25: Return bracket-paired sentence                                            ▷ Final output

---

