# OpenReview forum: "What does GPT store in its MLP weights? A case study of long-range dependencies"
_ICLR.cc/2024/Conference — ICLR 2024 Conference Withdrawn Submission_

### Official Review · Reviewer_cp3K · 2023-10-27

**Soundness:** 3 good
**Presentation:** 3 good
**Contribution:** 2 fair
**Rating:** 3
**Confidence:** 4

**Summary:**

The paper studies where a GPT2-family model stores information representing the state of a bracketing structure and how it is retrieved. The study is based on the method of activation patching of Meng et al. The authors demonstrate that the “bracketing” state is persistent in the very early MLP layers and investigate the way it is retrieved.

**Strengths:**

* To the best of my knowledge, the paper is the first to localise the “state” of the bracketing sequence and to try understand the underlying mechanism;
* The paper provides an interesting investigation on how the information is processed/retrieved.

**Weaknesses:**

* The title claims that the paper studies long-term dependencies, however, the actual experiments are done on very short inputs of 5 tokens. I understand there are technical difficulties related to studying longer sequences, however, I feel the title as-is misrepresents the content, so this should be resolved in either way.
* There is a considerable body of related work that seems to be extremely relevant yet it is not mentioned at all.
    * Studying behavior of neural nets on artificial languages (including Dyck languages) as a tool for gaining insight of their inner working and limitations, e.g. (Learning the Dyck Language with Attention-based Seq2Seq Models, Yu et al), (Feature Interactions Reveal Linguistic Structure in Language Models, Jumelet and Zuidema)
   * Studying long-range agreement in neural nets trained on natural languages. In particular (The emergence of number and syntax units in LSTM language models, Lakretz et al) managed to localize individual cells responsible for number agreement. Specifically this paper seems to employ a related method of suppressing activations of individual neurons* .

* In most of the experiments (except for the Appendix A), the paper studies only one instance of the GPT2-small, hence it is not given that the conclusions can be generalized.
* For many years, the BlackBox/neural net-interpretability domain was experimenting with (a) natural language long-distance dependencies, and (b) nested bracketing languages. Hence, solely focusing on non-nested bracketing strings of length 5 seems to be a very toy scenario in comparison.
* I wonder if the paper can provide any actionable take-away. For now the Conclusion mainly lists “supporting the hierarchical hypothesis”.

**Questions:**

I would appreciate addressing any of the weaknesses.

---

### Official Review · Reviewer_Un4f · 2023-10-29

**Soundness:** 4 excellent
**Presentation:** 3 good
**Contribution:** 3 good
**Rating:** 6
**Confidence:** 3

**Summary:**

The paper investigates where is the information stored in multi-layer
perceptrons in the case of bracketed sentences. That is, identify and
match closing brackets.
The authors use circuit analysis to understand how GPT-2 identifies
closing brackets. In particular, the authors use activation patching to
interactively trace the contributions of individual components in the
output distribution. They also use embedding projections, attention
analysis, and linear probes.

In this case, it is shown that low-level linguistic information, such
as brackets, is stored in the very early layers of the MLP. In summary,
increasingly complex information seems to be stored in progressively
MLP layers, while simple linguistic information are stored in the
first layers. They also found that the residual activation to predict
the closing bracket lies "dormant" and it is activated by later
layers.

**Strengths:**

- The authors performed a thorough study with different tests
- They show results which supports their claims
- The work represents a step forward towards understanding more DL
networks.

**Weaknesses:**

- The outcomes are not surprising as it is expected that simple
information is stored in the first layer and as the information flows
into deeper layers, more complex relationships are established
- In this sense, it is not very clear the significance of the reported
results.

**Questions:**

The authors focused mainly in short sequences and it is not completely
clear the behavior in much larger sequences.

It is not clear why the plots in Figure 7 are very different with
different type of brackets. I was expected to see similar curves for
all types of brackets.

---

### Official Review · Reviewer_5apn · 2023-11-01

**Soundness:** 2 fair
**Presentation:** 2 fair
**Contribution:** 2 fair
**Rating:** 3
**Confidence:** 2

**Summary:**

In this work, the authors study how information is stored and retrieved in GPT for tasks involving syntactic dependency of sentences. Specifically, they study the examples of bracketed sentences. The authors find that early layers are responsible for storing such syntactic information.

**Strengths:**

1) Authors study a simple "probing" task, that is easy to interpret
2) Findings make sense, and are interesting: finding that syntactic information is stored in earlier layers

**Weaknesses:**

1) Some of the rationale for the testing dataset used for analysis are not clear: e.g. why are n=5 sequences chosen? why is the open bracket token location fixed?
2) The authors do not compare with other methods of finding how information is encoded in LLMs: e.g. https://aclanthology.org/P19-1356.pdf. While not quite mechanistic interpretability, such techniques also provide a method to answer the questions the authors ask.
3) While the analysis about the residual output is interesting, it is quite short and does not seem complete to explain the phenomenon observed: can authors expand on this analysis further?

**Questions:**

1) While the analysis about the residual output is interesting, it is quite short and does not seem complete to explain the phenomenon observed: can authors expand on this analysis further?

---

### Meta-Review · Area_Chair_VsAJ · 2023-12-01

**Metareview:**

The paper investigates how the GPT2 language model stores and retrieves information by probing where the model stores the information necessary to keep track of opening and closing brackets. It finds that low MLP layers associated to the source position play a crucial role. Moreover, the first MLP layer contains distinct representations for different bracket types that can be manipulated to change the models' predictions.

The paper provides a new piece of evidence in the context of current attempts to decode a language model's functioning at the mechanistic level. The study is rather limited in that it entirely focuses on the GPT2 model and on a very specific formulation of the long-distance dependency task. However, it provides interesting insights that I find would be worth reporting at ICLR.

Unfortunately, the reviewers found the limitations to be too substantial, and the authors did not provide a rebuttal, so I would not feel comfortable overturning the reviewers' negative scores.

**Justification For Why Not Higher Score:**

Two reviewers over 3 provided a very negative scores, 1 a mildly positive one. As the authors did not provide a rebuttal, I felt there was no reason for the reviewers to revise their scores.

**Justification For Why Not Lower Score:**

N/A

---

### Decision · Program_Chairs · 2024-01-16

Reject